# Antimicrobial Activity of Dihydroisocoumarin Isolated from Wadi Lajab Sediment-Derived Fungus *Penicillium chrysogenum*: In Vitro and In Silico Study

**DOI:** 10.3390/molecules27113630

**Published:** 2022-06-06

**Authors:** Raha Orfali, Shagufta Perveen, Mohamed Fahad AlAjmI, Safina Ghaffar, Md Tabish Rehman, Abdullah R. AlanzI, Saja Bane Gamea, Mona Essa Khwayri

**Affiliations:** 1Department of Pharmacognosy, College of Pharmacy, King Saud University, P.O. Box 2457, Riyadh 11451, Saudi Arabia; malajmii@ksu.edu.sa (M.F.A.); sghafar.c@ksu.edu.sa (S.G.); mrehman@ksu.edu.sa (M.T.R.); aralonazi@ksu.edu.sa (A.R.A.); saja2996@gmail.com (S.B.G.); mona@ksu.edu.sa (M.E.K.); 2Department of Chemistry, School of Computer, Mathematical and Natural Sciences, Morgan State University, Baltimore, MD 21251, USA

**Keywords:** *Penicillium chrysogenum*, dihydroisocoumarin, microbial resistance, molecular modeling

## Abstract

Antibiotic resistance is considered a major health concern globally. It is a fact that the clinical need for new antibiotics was not achieved until now. One of the most commonly prescribed classes of antibiotics is β-Lactam antibiotics. However, most bacteria have developed resistance against β-Lactams by producing enzymes β-Lactamase or penicillinase. The discovery of new β-Lactamase inhibitors as new antibiotics or antibiotic adjuvants is essential to avoid future catastrophic pandemics. In this study, five dihydroisocoumarin: 6-methoxy mellein (1); 5,6-dihydroxymellein (2); 6-hydroxymellein (3); 4-chloro-6-hydroxymellein (4) and 4-chloro-5,6-di-hydroxymellein (5) were isolated from Wadi Lajab sediment-derived fungus *Penicillium chrysogenum*, located 15 km northwest of Jazan, KSA. The elucidation of the chemical structures of the isolated compounds was performed by analysis of their NMR, MS. Compounds **1**–**5** were tested for antibacterial activities against Gram-positive and Gram-negative bacteria. All of the compounds exhibited selective antibacterial activity against Gram-positive bacteria *Staphylococcus aureus* and *Bacillus licheniformis* except compound **3**. The chloro-dihydroisocoumarin derivative, compound **4**, showed potential antimicrobial activities against all of the tested strains with the MIC value between 0.8–5.3 μg/mL followed by compound **5**, which exhibited a moderate inhibitory effect. Molecular docking data showed good affinity with the isolated compounds to β-Lactamase enzymes of bacteria; NDM-1, CTX-M, OXA-48. This work provides an effective strategy for compounds to inhibit bacterial growth or overcome bacterial resistance.

## 1. Introduction

Discovery of antibiotics and the broad use of antibiotics play a clinically significant role in the treatment of bacterial infections, prolonging human life and reducing mortality. However, the rise of antibacterial resistance has caused a severe loss in the effectiveness of antibiotics [1,2]. The predominant mechanisms of bacterial resistance involve enzyme-mediated drug degradation, antibiotic efflux and lowered permeability of bacterial cell wall [3]. Although the dramatic rise of bacterial antibiotic-resistant poses a significant risk to morbidity and mortality globally, antimicrobial resistance (AMR) as one of the top 10 global public health threats facing humanity in 2019, pharmaceutical research and development have failed to fulfill the clinical requirement for novel antibiotics [4,5].

Indeed, the misuse of antibiotics is the main cause of the rising bacterial resistance to beta lactam antibiotics [6]. A class of metallo-β-Lactamases (MBLs), New Delhi metallo-β-Lactamases (NDM-1), was the main focus of researchers in recent years, because most of the β-Lactam antibiotics are losing activity, except monobactam aztreonam [7]. NDM-1 is mainly identified in *Acinetobacter* spp., *Escherichia coli* and in *Klebsiella pneumoniae* [8]. Research scientists reported that resistance of the gene *bla*_NDM-1_ was spreading widely all over the world. Presently, NDM has 17 variants, however clinically there is a lack of effective inhibitors of NDM-1. Pathogens overwhelming the *bla*_NDM-1_ gene potentially represent an imminent, tangible and challenging risk to human life or health. The discovery of new antibiotic adjuvants as NDM-1 inhibitors is crucial to prevent future disastrous pandemics [9]. Previous reports revealed that the CTX-M-type extended-spectrum β-Lactamases (ESBLs) were spreading rapidly and massively around the globe. Now, in Enterobacteriaceae, these enzymes are the utmost prevalent ESBLs and are also found infrequently in *Acinetobacter baumannii* and *Pseudomonas* spp. [10]. A class D carbapenemase, OXA-48, represents the main concern, due to issues with its findings and its link with the failure of treatments. Furthermore, variants of the OXA-48-like enzyme are plasmid-coded, and therefore linked with speedy spreading in public settings [11,12]. The Gram-negative bacteria that generally produce OXA-48-like enzymes are *E*. *coli*, *E. cloacae*, *S. xiamenensis*, *S. marcescens*, *C. freundii*, *K. oxytoca*, *K. pneumoniae*, *E. sakazakii*, *P. rettgeri* and *A. baumannii* [13].

From the time when Alexander Fleming discovered penicillin by the fungus *Penicillium notatum*, scientists were taking a deep interest in studying genus *Penicillium,* due to its capability of providing tremendously diverse secondary metabolites, and many of them possess pharmaceutical and biotechnological uses [14]. *P. chrysogenum* (renamed *P. rubens*) is the most investigated member of the greater than 354 *Penicillium* species in the same genus [15]. *Penicillium* is well-known for the capability of producing β-Lactam antibiotics, especially penicillin at an industrial scale, and the present strain is a result of the improvement of old classical strain [16,17]. *Penicillium chrysogenum* produces a variety of secondary metabolites with different activities, for example, siderophores, fungisporin (a hydrophobic cyclopeptide), roquefortines, ω-hydroxyemodin, pernitric acid, chrysogenin (a NRPS-derived yellow pigment), chrysogine, sorbicillinoids, and the sesquiterpene hydrocarbons and potentially capable of producing other compounds that were not identified before [14,18].

Dihydroisocoumarins (1*H*-2-benzopyran-1-ones or isocoumarin derivatives) are a group of structurally, pharmacologically and biosynthetically interesting secondary metabolites. Dihydroisocoumarins are isomers of coumarins that have a reversed α-pyrone ring moiety in their structures that might possess the 6,8-dioxygenated pattern, 3-(un) substituted phenyl ring or 3-alkyl chain (C_1_-C_17_) [19,20]. Substitution on the C-3 position of dihydroisocoumarin provides chemical diversity in synthetic and natural dihydroisocoumarin derivatives [21]. It also represents good intermediates for the biosynthesis of a wide variety of carbocyclic and heterocyclic natural products, such as isochromenes, isoquinolines and several aromatic compounds [22]. Several studies showed that the metabolites of isocoumarins have several bioactivities: cytotoxic; antimicrobial; algicidal; antimalarial; immunomodulatory; antiallergic; plant growth regulatory; protease inhibitors and acetylcholinesterase [23,24,25]. Mostly natural dihydroisocoumarins were isolated from various terrestrial and marine natural sources, such as plants, marine organisms, liverworts, insects, microbes, bacteria and fungi (endophytic, soil, and marine fungi) [26,27].

In this study, we report the details of isolation and elucidation of chemical structures of five dihydroisocoumarin derivatives: 6-methoxy mellein 1; 5,6-dihydroxymellein 2; 6-hydroxymellein 3; 4-chloro-6-hydroxymellein 4 and 4-chloro-5,6-di-hydroxymellein 5, for the first time isolated from *P. chrysogenum* obtained from the sediments of Wadi Lajab, located 15 km northwest of Jazan, KSA (Figure 1). The antimicrobial bioactivity potential against the following Gram-positive and Gram-negative bacteria was evaluated: *Pseudomonas aeruginosa* (NR-117678.1); *Staphylococcus aureus* (CP011526.1); *Enterobacter xiangfangensis* (CP017183.1)*; Bacillus licheniformis* (KX785171.1) and *Escherichia fergusonii* (CU928158.2). A molecular docking study is reported here, using the enzymes NDM-1, CTX-M-15 and OXA-48 as the possible target of β-Lactams antibiotics and investigating the structure activity relationships of the isolated compounds.

## 2. Results and Discussion

### 2.1. Chemistry

*P. chrysogenum* was isolated from the sediments of Wadi Lajab, located 15 km northwest of Jazan, KSA. *P. chrysogenum* was cultured on solid rice medium at 30 ± 2 °C and extracted with ethyl acetate to obtain a brownish residue. Repeated column chromatographic separations led to the isolation of the five dihydroisocoumarin derivatives: 6-methoxy mellein 1; 5,6-dihydroxymellein 2; 6-hydroxymellein 3; 4-chloro-6-hydroxymellein 4; 4-chloro-5,6-di-hydroxymellein 5. (Figure 1). Their structures were elucidated by mass spectrometry and nuclear magnetic resonance spectroscopy and further compared with literature data.

Mellein derivatives are a widely distributed dihydroisocoumarin, found in different kinds of fungi, plants and insects [28]. Based on the comparison of the obtained spectral data and literatures, compound **1** was identified as 6-methoxy mellein [29]. The 6-methoxy mellein was previously isolated from the culture broth of fungus *Sporormia affinis* and from plant *Daucus carota* (carrot) [30], but this is the first isolation of 6-methoxy mellein from the genus *Penicillium*. It was supposed that the production of this compound **1** caused the alteration in plant metabolism induced by the fungi [31], suggesting that 6-methoxymellein induces the active defenses of carrot-based compounds against fungi; thus, 6-methoxymellein was known as phytoalexin.

The 5,6-dihydroxymellein 2 was isolated as red powder, its ESIMS spectrum showed molecular ion signal at *m/z* 210. The ^13^C NMR spectrum showed the presence of three oxygenated aromatic carbons, one aromatic methine, five quaternary aromatic carbons, one aliphatic methine and one aliphatic methylene, in addition to cyclic ester carbonyl. All of the NMR and mass data match the published literature, which confirmed the structure of compound **2** as one of the mellein derivatives. The 5,6-dihydroxymellein 2 was isolated from spruce tree endophyte fungi *Cryptosporiopsis* sp., but this is the first report from *Penicillium*.

The 6-Hydroxymellein 3 was previously isolated from *Aspergillus terreu,* and it was found as a molecular investigation agent of anther and pollen development in higher plants, such as *Arabidopsis thaliana* [32]. The 6-Hydroxymellein biosynthetic gene cluster was also identified in the Lichen *Cladonia uncialis* and in *Daucus carota* (carrot) [33]. The 6-Methoxymellein 1 and 6-hydroxymellein 3 were found as biosynthetic precursors and isolated from the carrot root infected with the fungus *Fusarium solani* and *Sclerotium rolfsii*. Triggering of the synthesis of these compounds was also achieved in the carrot cell suspension cultures. The 6-Hydroxymellein 3 concentrate inside the carrots’ cells, while 6-methoxymellein 1 diffuses in the culture medium. Fungal secondary metabolites 1 and 3 have very similar structures. The mass spectrum of compound **3** showed an absence of methoxy methyl located at C-6 in the compound **1** [34]. All of the NMR and mass data match the published literature, which confirmed the structure of compound **3** as mellein derivative. Herein, we report for the first time the isolation of compound **3** from the genus *Penicillium*.

Compound **4**, 4-chloro-6-hydroxymellein, was isolated as red powder, its ESIMS spectrum showed a molecular ion signal at *m/z* 228. The ^13^C NMR spectra showed the presence of ten carbons, including one methyl, four methines and five quaternary carbons. The ^1^H NMR spectrum showed a high similarity to compound **3**, except the presence of an additional chloride at position 4 (δ_H_ 4.86, d, 2.0 Hz), instead of an aliphatic methylene group present in compound **3**. In addition, the downfield chemical shift at δ_H_ 4.60, dd, 2.0, 6.3 Hz for the methine proton at position 3 is in good agreement with the literature value given for the presence of vicinal halogen moiety. These data proposed compound **4** to be 4-chloro-6-hydroxymellein and the HMBC experiments further confirmed the structure. Comparing the spectral data with the published data, it was found that compound **4** was previously isolated from cultures of Ascomycete *Lachnum papyraceum* and tested for the nematicidal and antimicrobial activities. This is the first isolation of compound **4** from the genus, *Penicillium chrysogenum* [35].

Compound **5** was isolated as red powder, its ESIMS spectrum showed a molecular ion signal at *m/z* 224. The ^13^C NMR spectrum indicated the presence of three oxygenated aromatic carbons, one aromatic methine, five quaternary aromatic carbons and two aliphatic methines, in addition to cyclic ester carbonyl. The ^1^H NMR spectrum confirmed the presence of a tri-substituted aromatic ring, a halogenated methine proton and oxymethine carbon. Matching the spectroscopic data with the literature identified compound **5** as 4-chloro-5,6-di-hydroxymellein, which was previously isolated from the cultures of Ascomycete *Lachnum papyraceum* [36]. Herein, we report for the first time the isolation of compound **5** from the genus *P. chrysogenum*.

### 2.2. Biology

#### Bioactivity of Tested Compounds against the Pathogenic Bacteria

Compounds **1**–**5** were tested for their antibacterial activity against five pathogenic bacteria *Staphylococcus aureus* (CP011526.1), *Bacillus licheniformis* (KX785171.1), *Enterobacter xiangfangensis* (CP017183.1)*, Escherichia fergusonii* (CU928158.2) and *Pseudomonas aeruginosa* (NR-117678.1). Interestingly, all of the isolated compounds showed selective activity against Gram-positive bacteria *S. aureus* and *B. licheniformis*, except compound **3** (Table 1).

Compound **4** showed the most potent antibacterial activity against Gram-positive bacteria, with MIC 1.00 and 0.8 μg mL^−1^ for *S. aureus* and *B. licheniformis,* respectively. The presence of chlorine substitution at compound **4** improves the bioactivity in comparison with 4 and 5. Previous studies also validate the results of our study, the dichloro-dihydroisocoumarins compounds revealed antibacterial activity against *Bacillus subtilis*, *Staphylococcus aureus*, *Klebsiella pneumoniae*, *Escherichia coli* and *Acinetobacter calcoaceticus* [37].

### 2.3. Molecular Modelling

#### 2.3.1. Interaction between NDM-1 and Selected Ligands

The three-dimensional structure of NDM-1 is characterized by an αβ/βα four-layer fold, harboring a wide and deep substrate binding site. The most characteristic feature of the NDM-1 active site is the presence of two zinc ions: Zn1 coordinates with His120, His122 Asp124 and His189; Zn2 coordinates with Asp124, Cys208 and His250. Between the two zinc ions, a water molecule or hydroxide ion is placed between the two Zn ions, which act as a nucleophile during the hydrolysis of the β-Lactam ring in β-Lactam antibiotics. In addition to the active site residues, some other residues flanking the active site, such as Leu65, Met67, Pro68, Val73, Gly69, Phe70, Val73, Trp93, Leu209, Ile210, Lys211, Asp212, Lys214, Ala215, Lys216 and Asn220, were proposed to modulate the activity of NDM-1.

In this study, molecular docking between NDM-1 and selected ligands (compounds **1**–**5**) was performed and compared with the cognate ligand, meropenem. The results showed that all of the ligands bound to the active site of NDM-1 and occupied a position similar to that of meropenem (Figure 2A,B).

A close analysis of the NDM-1–meropenem interaction suggests that meropenem formed three hydrogen bonds with Gln123, His189 and Asn220, and a Pi-alkyl hydrophobic interaction with His250 (Figure 3A and Appendix A). In addition, several other residues such as Val73, Trp93, His122, Asp124, Ser217 and Gly219 formed Van der Waals’ intermolecular interaction with meropenem. It is worth noting that meropenem interacted with the key catalytic residues of NDM-1 such as His122, Asp124, His189 and His250, and some other residues, such as Val73 and Trp93, around the active site, which are reported to modulate NDM-1 catalytic activity. The docking-binding free energy and the binding affinity of meropenem towards NDM-1 were estimated to be −7.9 kcal mol^−1^ and 6.2 × 10^5^ M^−1^, respectively (Figure 3A).

An investigation of the interaction between NDM-1 and compound **1** revealed that compound **1** interacted with NDM-1 via two hydrogen bonds (Gln123 and Asp124) and five hydrophobic interactions (one with Phe70, and two with each residue, Val73 and Trp93). In addition, Van der Waals’ interactions were formed by His122, Cys208 and His250 (Figure 3B). Docking energy and the corresponding binding affinity of compound **1** for NDM-1 were measured to be −5.7 kcal mol^−1^ and 1.5 × 10^4^ M^−1^, respectively. These results confirmed that compound **1** interacts with catalytic residues (His122, Asp124, Cys208 and His250) of NDM-1, along with some other key residues such as Val73 and Trp93.

An analysis of the NDM-1–compound **2** interaction has shown that compound **2** interacted with NDM-1 through two hydrogen bonds (His120 and Asp124), an electrostatic interaction with Asp124 and a hydrophobic interaction with Val73. In addition, multiple Van der Waals’ interactions with compound **2** were formed by Trp93, Gln123, His122, His189, Cys208, Lys211, Asn220 and His250 (Figure 3C). Docking energy and the corresponding binding affinity of compound **2** for NDM-1 were determined to be −6.0 kcal mol^−1^ and 2.5 × 10^4^ M^−1^, respectively. These results confirmed that compound **2** interacts with catalytic residues (His120, His122, Asp124, His189, Cys208 and His250) of NDM-1 along with some other key residues, such as Val73, Trp93, and Asn220.

An analysis of the NDM-1–compound **3** interaction has shown that compound **3** interacted with NDM-1 through three conventional hydrogen bonds (His120, His189 and His250), one Pi-donor hydrogen bond with His250, an electrostatic interaction with Asp124 and four hydrophobic interactions with (two with Trp93, and one with Leu65 and Cys208 each). In addition, the Van der Waals’ interactions with compound **3** were formed by Val73, His122, and Asn220, (Figure 3D). Docking energy and the corresponding binding affinity of compound **3** for NDM-1 were determined to be −6.0 kcal mol^−1^ and 2.5 × 10^4^ M^−1^, respectively. These results confirmed that compound **3** interacts with catalytic residues (His120, His122, Asp124, His189, Cys208 and His250) of NDM-1 along with some other key residues, such as Val73, Trp93, and Asn220.

An investigation on the interaction between NDM-1 and compound **4** revealed that compound **4** interacted with NDM-1 through two hydrogen bonds (His120 and Asp124), an electrostatic interaction with Asp124 and four hydrophobic interactions (two with Trp93 and one with each residue Leu65 and Cys208). In addition, Van der Waals’ interactions were formed between compound **4** and Val73, His122, Gln123, His189, Asn220 and His250 (Figure 3E and Appendix A). Docking energy and the corresponding binding affinity of compound **4** for NDM-1 were determined to be −6.2 kcal mol^−1^ and 3.5 × 10^4^ M^−1^, respectively. These results confirmed that compound **4** interacts with catalytic residues (His120, His122, Asp124, His189, Cys208 and His250) of NDM-1 along with some other key residues, such as Val73 and Trp93.

An analysis of the NDM-1–compound **5** interaction has shown that compound **5** interacted with NDM-1 through three conventional hydrogen bonds (His120, His189 and His250), a carbon hydrogen bond with His189, an electrostatic interaction with Asp124 and three hydrophobic interactions (two with Val73 and one with His250). In addition, Van der Waals’ interactions were formed between compound **5** and Trp93, Gln123, His122, Cys208, Lys211 and Asn220 (Figure 3F). Docking energy and the corresponding binding affinity of compound **5** for NDM-1 were determined to be −6.2 kcal mol^−1^ and 3.5 × 10^4^ M^−1^, respectively. These results confirmed that compound **5** interacts with catalytic residues (His120, His122, Asp124, His189, Cys208 and His250) of NDM-1, along with some other key residues such as Val73, Trp93, and Asn220.

#### 2.3.2. Interaction between CTX-M-15 and Selected Ligands

The three-dimensional structure of CTX-M-15 is characterized by an αβα sandwich fold, harboring a wide and deep substrate binding site. The amino acid residues such as Ser70, Lys73, Asn104, Ser130, Glu166, Asn170, Lys234, Thr235, Ser237 and Arg274 play a crucial role in the acylation and deacylation of β-Lactam antibiotics, and hence modulate the activity of CTX-M-15 during hydrolysis.

In this study, molecular docking between CTX-M-15 and selected ligands (compounds **1**–**5**) was performed and compared with the cognate ligand, avibactam. The obtained results showed that all of the ligands were bound to the active site of CTX-M-15, and occupied a position similar to that of avibactam (Figure 2C,D). A close analysis of CTX-M-15–avibactam interaction suggests that avibactam formed three hydrogen bonds with Asn104, Thr216 and Lys234 (Figure 4A). Additionally, several residues, such as Ser70, Lys73, Tyr105, Ser130, Asn132, Asn170, Thr235, Gly236 and Ser237, formed a Van der Waals’ interaction with avibactam. It is worth noting that avibactam interacted with the key residues of CTX-M-15 such as Ser70, Lys73, Asn104, Ser130, Asn170, Lys234, Thr235 and Ser237. The docking energy and binding affinity of avibactam towards CTX-M-15 were estimated to be −6.8 kcal mol^−1^ and 9.7 × 10^4^ M^−1^, respectively (Figure 4A).

An investigation of the interaction between CTX-M-15 and compound **1** revealed that compound **1** interacted with CTX-M-15 through three hydrogen bonds with Ser70, Asn132 and Asn170. In addition, Van der Waals’ interactions were formed between compound **1** and Lys73, Asn104, Tyr105, Ser130, Glu166, Lys234, Ser237 and Gly238 (Figure 4B). These results confirmed that compound **1** interacts with the catalytic residues of CTX-M-15, such as Ser70, Lys73, Asn104, Ser130, Glu166, Asn170, Lys234 and Ser237. Docking energy and the corresponding binding affinity of compound **1** for CTX-M-15 were determined to be −6.7 kcal mol^−1^ and 8.2 × 10^4^ M^−1^, respectively.

An analysis of the CTX-M-15–compound **2** interaction showed that compound **2** interacted with CTX-M-15 through five hydrogen bonds (one each with residue Ser70, Ser130 and Glu166, and two with Asn132) and two hydrophobic interactions with Tyr105. In addition, Van der Waals’ interactions were formed between compound **2** by Lys73, Asn104, Asn170, Ser237 and Gly238 (Figure 4C and Appendix A). Docking energy and the corresponding binding affinity of compound **2** for CTX-M-15 were determined to be −6.5 kcal mol^−1^ and 5.8 × 10^4^ M^−1^, respectively. These results confirmed that compound **2** interacts with key residues of CTX-M-15, such as Ser70, Lys73, Asn104, Ser130, Glu166, Asn170, Ser237 and Arg274.

An analysis of the CTX-M-15–compound **3** interactions has shown that compound **3** interacted with CTX-M-15 through five hydrogen bonds with Ser70, Ser130, Asn132 (two interactions) and Asn170. In addition, some other residues, such as Lys73, Asn104, Tyr105, Glu166, Asn170, Ser237 and Gly238, formed Van der Waals’ interactions (Figure 4D). Docking energy and the corresponding binding affinity of compound **3** for CTX-M-15 were determined to be −6.5 kcal mol^−1^ and 5.8 × 10^4^ M^−1^, respectively. These results confirmed that compound **3** interacts with the key residues of CTX-M-15, such as Ser70, Lys73, Asn104, Ser130, Glu166, Asn170 and Ser237.

An investigation on the interaction between CTX-M-15 and compound **4** revealed that compound **4** interacted with CTX-M-15 through five hydrogen bonds with Lys73, Ser130, Asn132 (two interactions) and Asn170. In addition, some of the other residues such as Ser70, Asn104, Tyr105, Glu166, Ser237 and Gly238 formed Van der Waals’ interactions (Figure 4E). These results confirmed that compound **4** interacts with the catalytic residues of CTX-M-15 such as Ser70, Lys73, Asn104, Ser130, Glu166, Asn170 and Ser237. Docking energy and the corresponding binding affinity of compound **4** for CTX-M-15 were determined to be −6.8 kcal mol^−1^ and 9.7 × 10^4^ M^−1^, respectively.

An analysis of the CTX-M-15–compound **5** interaction has shown that compound **5** interacted with CTX-M-15 through five hydrogen bonds with Ser130 (two interactions), Asn132 (two interactions) and Asn170. In addition, some other residues, such as Ser70, Lys73, Asn104, Tyr105, Glu166, Ser237 and Gly238, formed Van der Waals’ interactions (Figure 4F). Docking energy and the corresponding binding affinity of compound **5** for CTX-M-15 were determined to be −6.7 kcal mol^−1^ and 8.2 × 10^4^ M^−1^, respectively. These results confirmed that compound **5** interacts with key residues of CTX-M-15, such as Ser70, Lys73, Asn104, Ser130, Glu166, Asn170 and Ser237.

#### 2.3.3. Interaction between OXA-48 and Selected Ligands

The three-dimensional structure of OXA-48 is characterized by a αβ sandwich fold, harboring a wide and deep active site. The amino acid residues, such as Ser70, Lys73, Ile102, Ser118, Val120, Thr209, Tyr211, Trp157, Leu247, Gly248 and Arg250, play a crucial role in the acylation and deacylation of the β-Lactam ring in β-Lactam antibiotics, and hence modulate the activity of CTX-M-15 during hydrolysis of the β-Lactam ring.

In this study, molecular docking between OXA-48 and selected ligands (compounds **1**–**5**) was performed and compared with the cognate ligand, imipenem. The results revealed that all of the ligands were bound to the active site of OXA-48 and occupied a position similar to that of imipenem (Figure 2E,F). A close analysis of OXA-48–imipenem interaction suggests that imipenem formed six hydrogen bonds with Ser70, Lys116, Thr209 (two interactions), Tyr211 and Arg250 (Figure 5A). In addition, several other residues, such as Ala69, Lys73, Ile102, Trp105, Met115, Ser118, Val120, Lys208, Gly210, Thr213 and Leu158, formed Van der Waals’ interactions with avibactam. It is worth noting that imipenem interacted with the key residues of OXA-48, such as Ser70, Lys73, Ile102, Ser118, Val120, Thr209, Tyr211 and Arg250. The docking energy and binding affinity of imipenem towards OXA-48 were estimated to be −6.1 kcal mol^−1^ and 2.9 × 10^4^ M^−1^, respectively (Figure 5A).

An analysis of the OXA-48–compound **1** interaction has shown that compound **1** interacted with OXA-48 through a single hydrogen bond with Tyr211, and three hydrophobic interactions with Ile102, Val120 and Tyr211. In addition, some other residues such as Ala69, Ser70, Trp105, Leu158, Gly210 and Arg214 formed Van der Waals’ interactions (Figure 5B).

Docking energy and the corresponding binding affinity of compound **1** for OXA-48 were determined to be −6.3 kcal mol^−1^ and 4.1 × 10^4^ M^−1^, respectively. These results confirmed that compound **1** interacts with the key residues of OXA-48, such as Ser70, Ile102, Val120 and Tyr211. An investigation of the interaction between OXA-48 and compound **2** revealed that compound **2** interacted with OXA-48 through three hydrogen bonds, one with Trp105 and two with Arg250. In addition, some other residues, such as Ser70, Ile102, Asn104, Tyr117, Ser118, Val120, Thr209, Gly210 and Tyr211, formed Van der Waals’ interactions with compound **2** (Figure 5C). These results confirmed that compound **2** interacts with catalytic residues of OXA-48, such as Ser70, Ile102, Ser118, Val120, Thr209, Tyr211, and Arg250. Docking energy and the corresponding binding affinity of compound **2** for OXA-48 were determined to be −6.3 kcal mol^−1^ and 4.1 × 10^4^ M^−1^, respectively.

An analysis of the OXA-48–compound **3** interaction has shown that compound **3** interacted with OXA-48 through two hydrogen bonds with Tyr211, and four hydrophobic interactions with Ile102, Val120, Tyr211 and Leu158. In addition, some other residues, such as Ala69, Ser70, Trp105 and Arg214, formed Van der Waals’ interactions with compound **3** (Figure 3D). Docking energy and the corresponding binding affinity of compound **3** for OXA-48 were determined to be −6.2 kcal mol^−1^ and 3.5 × 10^4^ M^−1^, respectively. These results confirmed that compound **3** interacts with the key residues of OXA-48, such as Ser70, Ile102, Val120, and Tyr211.

An investigation of the interaction between OXA-48 and compound **4** revealed that compound **4** interacted with OXA-48 through three hydrogen bonds (Thr209, Tyr211 and Arg250) and four hydrophobic interactions (three with Tyr211 and one with Leu247). In addition, some other residues, such as Ser70, Ile102, Ser118, Lys208 and Gly210, formed Van der Waals’ interactions with compound **4** (Figure 5E). These results confirmed that compound **4** interacts with catalytic residues of OXA-48, such as Ser70, Ile102, Ser118, Thr209, Tyr211, Leu247 and Arg250. Docking energy and the corresponding binding affinity of compound **4** for OXA-48 were determined to be −6.4 kcal mol^−1^ and 4.9 × 10^4^ M^−1^, respectively.

An investigation of the interaction between OXA-48 and compound **5** revealed that compound **5** interacted with OXA-48 through two hydrogen bonds (Tyr211 and Arg214) and six hydrophobic interactions (two with Ile102, two with Tyr211 and one each with Val120 and Leu158). In addition, some other residues, such as Ala69, Ser70, Trp105 and Gly210, formed Van der Waals’ interactions (Figure 5F). These results confirmed that compound **5** interacts with catalytic residues of OXA-48, such as Ser70, Ile102, Val120 and Tyr211. Docking energy and the corresponding binding affinity of compound **5** for OXA-48 were determined to be −6.2 kcal mol^−1^ and 3.5 × 10^4^ M^−1^, respectively.

## 3. Materials and Methods

### 3.1. General Experimental Procedures

The experimental procedures used in this study were reported previously [38,39]. The ^1^H, ^13^C NMR and 2D NMR spectra were recorded on a Bruker AMX-700 spectrometer with tetramethylsilane (TMS) as an internal standard. Chemical shifts are in ppm (δ), relative to tetramethylsilane as an internal standard and scalar coupling constants (J) reported in Hertz. ESI-MS analyses were measured on an Agilent Triple Quadrupole 6410 QQQ LC/MS mass spectrometer with ESI ion source (gas temperature is 350 °C, nebulizer pressure is 60 psi and gas flow rate is 12 L/min), operating in the negative and positive scan modes of ionization through the direct infusion method using CH_3_OH\H_2_O (1:1 *v*/*v*) at a flow rate of 0.2 mL/min. Column chromatography was carried out on silica gel and sephadex LH-20 (E. Merck, Darmstadt, Germany). Thin layer chromatography (TLC) was performed on precoated TLC plates (aluminum sheets, silica gel and RP-18 F254, Merck, Germany); the detection was completed at 254 nm and by spraying with ceric sulphate reagent. HPLC analysis was performed on a Prominence Shimadzu LC Solution, (Kyoto, Japan) and the system was equipped with a CBM-20A communication bus module, two LC-10AD pumps, a CTO-10A(C) column oven and an SPD 10A(V) diode array detector. A Shim-pack VP-ODS (150 mm × 4.6 mm, 5.0 μm, Shimadzu) analytical column was used and kept at 40 °C. The mobile phase consisted of water containing 0.1% trifluoracetic acid (A) and CH_3_OH (B). The flow rate was set at 0.4 mL/min and the injection volume was 20 μL. The DAD detection was achieved in the range of 254 nm.

### 3.2. Fungal Strain Materials

The fungal strain was isolated from the sediment of Wadi Lajab, located 15 km northwest of Jazan, KSA, in September 2019 and submitted to the Pharmacognosy department lab, KSU. The fungal strain was recognized as *Penicillium chrysogenum* (GenBank accession No. MH127462), according to DNA amplification sequencing of the fungal ITS region, as reported before [38,39,40,41].

### 3.3. Fermentation, Extraction and Isolation

The strain (*Penicillium chrysogenum*) was cultured on dried rice agar medium ASS, which was prepared by the autoclaving of commercially available milk rice (100 g) and 100 mL of water in a 1 L glass flask. The flasks were already sterilized by autoclaving at 120 °C for 25 min and then allowed to cool down to 27 °C. The strain RO-19-5-6-4-1 was cultivated in a constant and monitored temperature incubator at 25 °C under static conditions with shaking (170 rpm). The ethyl acetate extract of ASL (75 mg) was harvested at 15 d and ASS (100 mg) was harvested at 22 d, both extracts were subjected to antimicrobial and UPLC analysis. After assessment of the reported data, the strain was furtherly cultivated on rice media and fermented in twenty 1L flasks under the same previous conditions. After 20 days, full fungal growth was assessed and each flask was extracted with ethyl acetate (2 × 500 mL), followed by filtration and evaporation of each flask. The obtained crude extract (7 g) was partitioned between *n*-hexane and 95% aqueous MeOH. The MeOH extract was then subjected to chromatography on an open glass silica gel column (CC 400 mm × 50 mm) by using a gradient elution solvent system of *n*-hexane-ethyl acetate (90:10 to 50:50) to obtain six major fractions F_1_-F_6_. Fraction F_1_ was loaded on silica gel column (250 mm × 10 mm) and eluted with *n*-hexane-ethyl acetate (9:1) to give compound **1** (5.0 mg). Fraction F_3_ was eluted by using *n*-hexane-ethyl acetate (8.0:2.0), conducted to the isolation of compounds **2** and 5 (10 mg and 5 mg, respectively).

The sub-fraction F_4_, which was eluted with *n*-hexane–ethyl acetate (7.0:3.0) was a mixture of two compounds with few impurities. It was further subjected to open silica gel column (250 mm × 10 mm) using *n*-hexane-ethyl acetate (6.5:3.5) as eluent to afford 3 (15 mg) and 4 (12 mg).

(*S*)-6-Methoxy mellein 1

White amorphous powder, [α]^D^_25_ + 30.0 (*c* = 1.30 in CH_3_OH), HREIMS *m/z*: 208.02, ^1^H-NMR (DMSO, 700 MHz): δ_H_ 1.41 (3H, d, *J* = 6.3 Hz, CH_3_), 2.81 (1H, dd, *J* = 11.5, 16.0 Hz, H-4a), 2.90 (1H, dd, *J* = 3.0, 16.0 Hz, H-4b), 3.80 (3H, s, OCH_3_), 4.70 (1H, m, H-3), 6.30 (1H, d, 2.2 Hz, H-7), 6.36 (1H, d, 2.2 Hz, H-5), 10.95 (8-OH). ^13^C-NMR (DMSO, 125 MHz): δ_c_ 18.6 (CH_3_), 34.2 (C-4), 56.0 (OCH_3_), 75.5 (C-3), 100.1 (C-9), 101.3 (C-7), 107.1 (C-5), 142.5 (C-10), 164.5 (C-6), 165.2 (C-8), 169.2 (C-1).

(*S*)-5,6-Dihydroxymellein 2

White amorphous powder, [α]^D^_25_ + 60.0 (*c* = 0.06 in CHCl_3_), HREIMS *m/z*: 210.03, ^1^H-NMR (DMSO, 700 MHz): δ_H_ 1.45 (3H, d, *J* = 6.3 Hz, CH_3_), 2.40 (1H, dd, *J* = 11.0, 16.2 Hz, H-4a), 2.98 (1H, dd, *J* = 3.0, 16.2 Hz, H-4b), 4.54 (1H, m, H-3), 6.20 (1H, s, H-7), 11.0 (8-OH). ^13^C-NMR (DMSO, 125 MHz): δ_c_ 19.9 (CH_3_), 29.5 (C-4), 75.0 (C-3), 100.1 (C-9), 102.3 (C-7), 126.2 (C-5), 135.6 (C-10), 155.2 (C-6), 158.1 (C-8), 169.1 (C-1).

(*S*)-6-Hydroxymellein 3

White amorphous powder, [α]^D^_25_ − 51.0 (*c* = 0.10 in CH_3_OH), HREIMS *m/z*: 194.02, ^1^H-NMR (DMSO, 700 MHz): δ_H_ 1.39 (3H, d, *J* = 6.6 Hz, CH_3_), 2.82 (1H, dd, *J* = 11.2, 16.1 Hz, H-4a), 2.92 (1H, dd, *J* = 2.8, 16.1 Hz, H-4b), 4.68 (1H, ddd, 2.8, 5.6, 11.2 Hz, H-3), 6.20 (1H, d, 2.1 Hz, H-7), 6.24 (1H, d, 2.1 Hz, H-5), 11.14 (8-OH). ^13^C-NMR (DMSO, 125 MHz): δ_c_ 20.7 (CH_3_), 34.1 (C-4), 75.8 (C-3), 100.5 (C-9), 101.3 (C-7), 107.2 (C-5), 142.2 (C-10), 163.8 (C-6), 164.9 (C-8), 169.9 (C-1).

(*R*)-4-Chloro-6-hydroxymellein 4

White amorphous powder, [α]^D^_25_ − 74.5 (*c* = 1.05 in CHCl_3_), HREIMS *m/z*: 228.01, ^1^H-NMR (DMSO, 700 MHz): δ_H_ 1.41 (3H, d, *J* = 6.3 Hz, CH_3_), 4.86 (1H, dd, *J* = 2.0, 6.3 Hz, H-3), 4.60 (1H, d, 2.0 Hz, H-4), 6.35 (1H, d, 2.0 Hz, H-7), 6.55 (1H, d, 2.0 Hz, H-5), 11.05 (8-OH). ^13^C-NMR (DMSO, 125 MHz): δ_c_ 17.5 (CH_3_), 57.2 (C-4), 75.2 (C-3), 100.0 (C-9), 102.4 (C-7), 108.0 (C-5), 141.5 (C-10), 163.1 (C-6), 165.0 (C-8), 168.1 (C-1).

(*R*)-4-Chloro-5,6-di-hydroxymellein 5

White amorphous powder, [α]^D^_25_ − 5.8 (*c* = 0.4 in CHCl_3_), HREIMS *m/z*: 244.42, ^1^H-NMR (DMSO, 700 MHz): δ_H_ 1.44 (3H, d, *J* = 6.3 Hz, CH_3_), 4.85 (1H, m, H-3), 4.65 (1H, d, 2.2 Hz, H-4), 6.43 (1H, s, H-7), 11.21 (8-OH). ^13^C-NMR (DMSO, 125 MHz): δ_c_ 19.1 (CH_3_), 56.8 (C-4), 76.0 (C-3), 101.4 (C-9), 101.5 (C-7), 136.5 (C-5), 141.4 (C-10), 156.2 (C-6), 158.3 (C-8), 168.9 (C-1).

### 3.4. Antibacterial Assay 

#### 3.4.1. Agar Diffusion Method

The antibacterial activity of the isolate was evaluated according to the reported method [42]. Bacterial strains were grown in a nutrient broth for 24 h and then spread on Mueller–Hinton agar plate. A total of 10 µL of the sample solution were loaded in wells, using Amikacin as a positive control. The clear area, which was free of microbial growth, was measured in triplicate, the diameter of the inhibition zone and the mean of the measured diameters were recorded.

#### 3.4.2. Minimum Inhibitory Concentration

The minimum inhibitory concentrations (MIC) of compounds **1**–**5** were determined by broth microdilution technique, as described by Ebrahim W., et al. [43]. Compounds **1**–**5** were separately dissolved in less than 10% dimethyl-sulfoxide (DMSO)/Mueller–Hinton Broth (MHB). A total of 100 μL of MHB were dispensed in each well of a sterile 96 well plate. In order to achieve the desired concentration, an appropriate volume of each compound was added to the first well and serially diluted twofold in a 96-well microplate. The highest concentration was 25 μg mL^−1^ for each compound. Amikacin was used as a reference drug. Positive and negative control wells were used. The negative control wells consisted of MHB and the positive control contained bacterial suspension without any compounds. For the bacterial inoculate, the bacterial strains were streaked onto Mueller–Hinton agar and, after 18–24 h incubation at 37 °C, three to five morphologically identical colonies were picked and transferred into phosphate-buffered saline (PBS) and diluted to 0.5 McFarland standard. The obtained suspension was further diluted 1:100 in media. The bacterial suspension was added to the test plates to get a final concentration 5 × 10^5^ CFU/ml. The plates were incubated for 24 h at 37 °C. Results were visually checked for turbidity, and the MIC values were the lowest concentration where growth was not visible. The MIC value of each sample, defined as the lowest sample concentration that inhibited complete bacteria growth, was detected by the addition of 0.2 mg/mL 2-[4-iodophenyl]-3-[4-dinitrophenyl]-5-phenyl-tetrazolium chloride (INT) and incubated at 37 °C for 30 min. Viable bacteria reduced the yellow dye to pink.

### 3.5. Molecular Docking

Molecular docking between the target proteins (NDM-1, CTX-M-15 and OXA-48) and selected ligands (compounds **1**–**5**) was performed using AutoDock4.2, as reported previously Morris et al. [44] and Rehman et al. [45]. Briefly, the two-dimensional structures of ligands (compounds **1**–**5**) were drawn in ChemSketch and saved as a mol2 file. The preprocessing of the ligands was performed by adding Gasteiger partial charges, merging non-polar H-atoms and defining rotatable bonds, as reported earlier [46,47]. The three-dimensional coordinates of the target proteins were downloaded from the PDB-RCSB database. The X-ray crystal structure of NDM-1 (PDB Id: 4EYL) bound with hydrolyzed meropenem was solved to a resolution of 1.9 Å [47], while the crystal structures of CTX-M-15 bound with inhibitor avibactam (PDB Id: 4HBU) and OXA-48 bound with imipenem (PDB Id: 6P97) were solved to a resolution of 1.1 and 1.8 Å, respectively [47]. Before molecular docking, the targeted proteins were preprocessed by eliminating their bound ligands and any other heterogeneous molecules. Non-essential water molecules were also deleted from the structure files, and any missing side chains and loops were modeled. AutoDock tools (ADT) were used to add essential H-atoms, Kollman’s united-atom-type charges and solvation parameters. The energy of the whole molecule was minimized using the Charmm36 forcefield. Molecular docking of the ligands with NDM-1, CTX-M-15 and OXA-48 was performed inside a grid box of dimension 20.1 × 19.8 × 20.7 Å centered at 8.1 × −39.9 × 5.5 Å, 17.6 × 26.8 × 16.6 Å centered at −7.9 × −2.1 × 14.0 Å and 19.1 × 15.8 × 19.9 Å centered at 7.1 × −21.3 × 2.1 Å, respectively, with a spacing of 0.375 Å using the AutoGrid program. Lamarck Genetic Algorithm (LGA), along with the Solis and Wets local search method, were applied to perform molecular docking. All of the parameters were set to their default values and eight docking runs were performed. A maximum of 2,500,000 energy calculations was computed for each docking run. The docking pose with a minimum RMSD (Root Mean Square Deviation) value was selected for the experiment in Discovery Studio-2016 (Accelrys).

## 4. Conclusions

For many years, the media and the scientific community have published many fearful articles regarding the rise of antibiotic resistance, even citing treatment failure in multidrug-resistant bacteria (MDR)-infected patients which results in high mortality worldwide. This caused conjectures about the threat of multidrug-resistant bacteria toward the modern world, because of both a global rise in bacterial resistance and the lack of discovery of new compounds to treat MDR bacterial infections. Increased antibiotic resistance and an absence of new antibiotic development showed the need for finding new compounds. Despite the progress in new antibiotic development research, the main bottleneck in detecting potentially interesting compounds, which may be used to overcome bacterial resistance and the treatment of infections, was not yet solved. Interestingly, dihydroisocoumarin compounds isolated from the Wadi Lajab sediment fungus, *P. chrysogenum,* showed selective antibacterial activity against Gram-positive bacteria *S. aureus* and *B. licheniformis*, except for compound **3**. In compounds **4** and **5**, the presence of the chlorine group at C-4 improved the activity. The molecular modeling study revealed good affinity of the dihydroisocoumarin-selected ligands (compounds **1**–**5**) with the possible target proteins (NDM-1, CTX-M-15 and OXA-48); these compounds interacted with β-Lactamase enzymes. Thus, dihydroisocoumarins could be employed for antibacterial compound studies in the future.

## Figures and Tables

**Figure 1 molecules-27-03630-f001:**
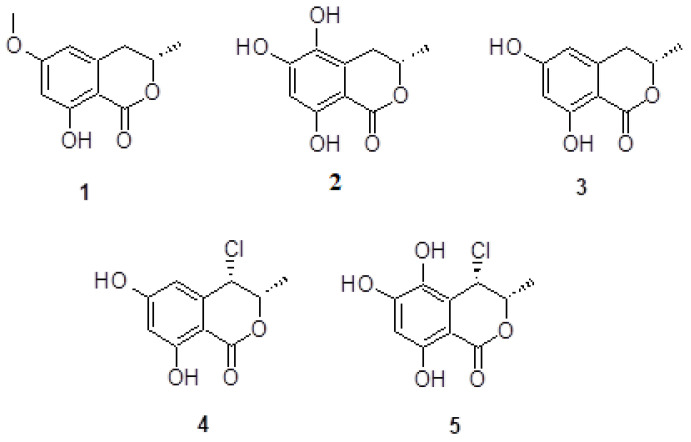
Chemical structures of compounds **1**–**5**.

**Figure 2 molecules-27-03630-f002:**
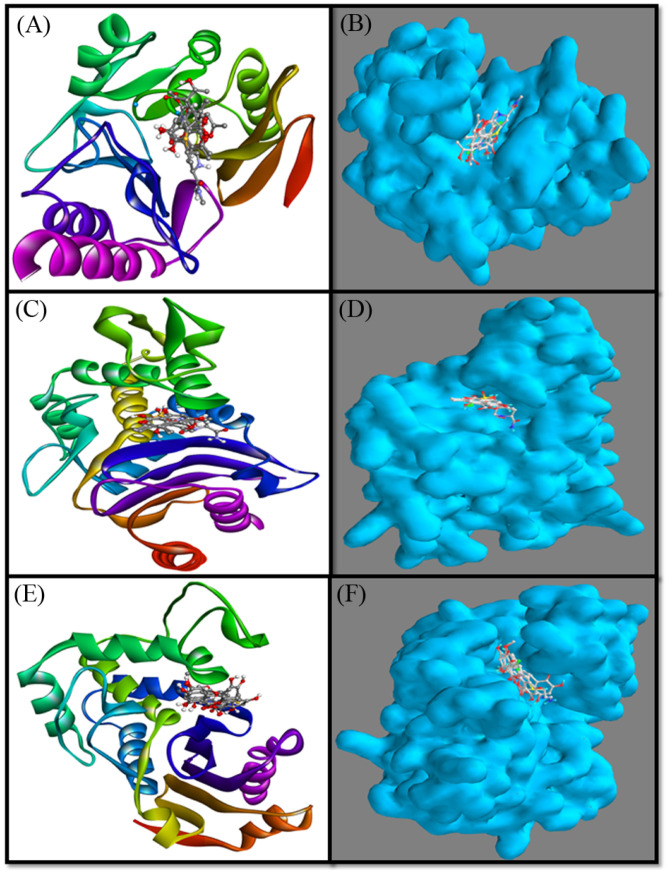
Cartoon and Van der Waals surface representation of meropenem (**A**,**B**), respectively, inside the active site of NDM-1; avibactam (**C**,**D**), respectively, inside the active site of CTX-M-15; and imipenem (**E**,**F**), respectively, inside the active site of OXA-48.

**Figure 3 molecules-27-03630-f003:**
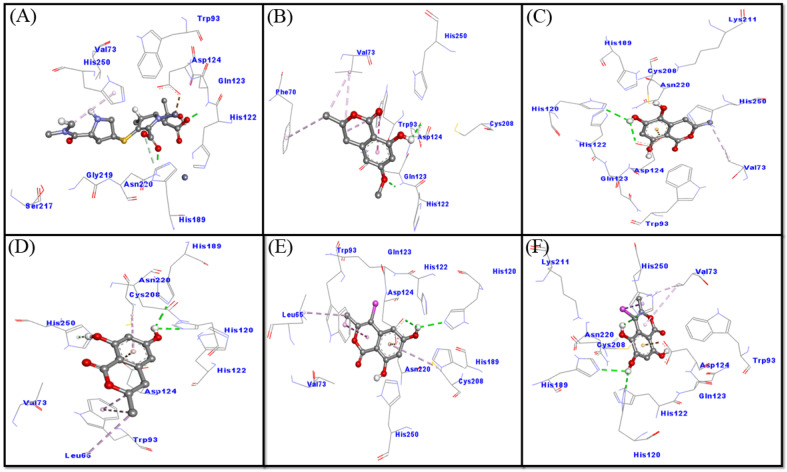
Binding mode of the ligands: meropenem (**A**); compound **1** (**B**); compound **2** (**C**); compound **3** (**D**); and compound **4** (**E**) and compound **5** (**F**) inside the active site of NDM-1.

**Figure 4 molecules-27-03630-f004:**
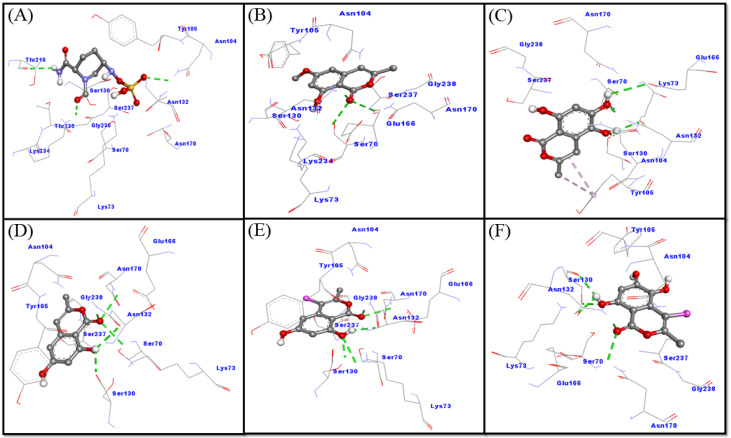
Binding mode of the ligands: avibactam (**A**); Compound **1** (**B**); compound **2** (**C**); compound **3** (**D**); and compound **4** (**E**) and compound **5** (**F**) inside the active site of CTX-M-15.

**Figure 5 molecules-27-03630-f005:**
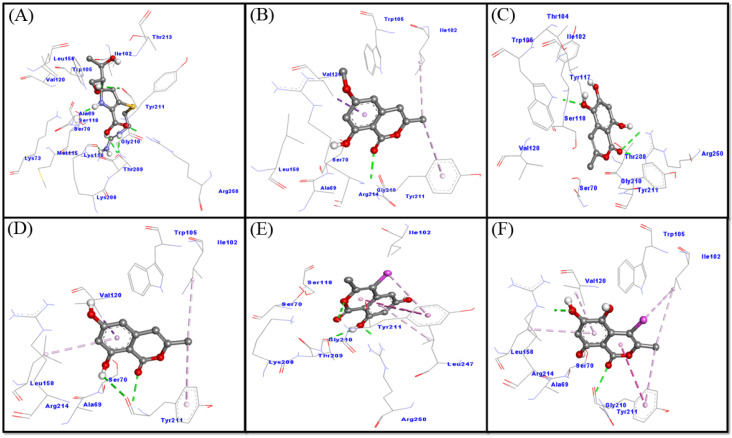
Binding mode of the ligands: imipenem (**A**); Compound **1** (**B**); compound **2** (**C**); compound **3** (**D**); and compound **4** (**E**) and compound **5** (**F**) inside the active site of OXA-48.

**Table 1 molecules-27-03630-t001:** Antibacterial Activity of Compounds **1**–**5** Expressed as MIC (μg/mL) *.

Compound	*S. aureus*	*B. licheniformis*	*E. xiangfangensis*	*E. fergusonii*	*P. aeruginosa*
**1**	10.4	12.2	21.6	18.4	14.8
**2**	14.5	16.3	>25	>25	>25
**3**	>25	>25	>25	>25	>25
**4**	1.00	0.8	3.5	4.7	5.3
**5**	3.8	4.3	5.2	7.5	9.7
Amikacin	0.5	0.2	0.3	0.4	0.8

* Results expressed as mean; each experiment was repeated three times.

## Data Availability

Not applicable.

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
