# Peer review of "Antimicrobial Activity of Dihydroisocoumarin Isolated from Wadi Lajab Sediment-Derived Fungus Penicillium chrysogenum: In Vitro and In Silico Study"

_molecules, 2022, doi:10.3390/molecules27113630_

Round 1

Reviewer 1 Report

In the current manuscript, the authors reported the isolation, antibacterial activity, docking of five isocoumarine analogues from Penicillium chrysogenum. Some of the compounds showed antibacterial activity.

There are some questions should be addressed.

  1. The absolute of the compounds should be elucidated.
  2. From the methods, the MICs were difficult to be determined. More details should added to show how the MICs were so difference from each other sample.
  3. From the docking energy, the results are not good enough to show good affinity.
  4. Page 1, abstract, line 18-20: put the compounds number to brackets.
  5. Page 1, line 21: Penicillium chrysogenum, it should be italic.
  6. Page 1, line 24: bacteria S. aureus and B. licheniformis, when they were mentioned the first time, should give the full name and use italic format.
  7. Page 3, Figure 1: please label the C numbers.
  8. Page 17, line 481: check the coupling model of H-3.

Author Response

Reviewer #1:

Comment

Response

1-The absolute of the compounds should be elucidated.

We mentioned the [a]D25 values of each compound and highlighted.

2-From the methods, the MICs were difficult to be determined. More details should added to show how the MICs were so difference from each other sample.

Correction done and highlighted in yellow.

3- From the docking energy, the results are not good enough to show good affinity.

In our study, we observed that the docking results are comparable to the control ligands. Hence, they can serve as scaffold to design more potent inhibitors of the studied beta-lactamases.

4-Page 1, abstract, line 18-20: put the compounds number to brackets.

Correction done and highlighted in yellow.

5-Page 1, line 21: Penicillium chrysogenum, it should be italic.

Correction done and highlighted in yellow.

6-Page 1, line 24: bacteria S. aureus and B. licheniformis, when they were mentioned the first time, should give the full name and use italic format.

Correction done and highlighted in yellow.

7-Page 3, Figure 1: please label the C numbers.

Correction done and highlighted in yellow.

.

8-Page 17, line 481: check the coupling model of H-3.

Corrected

Reviewer 2 Report

In the present study, the authors investigated antimicrobial activities of isocumarine and its derivatives based on in vitro and in silico studies. Bioactivity and interaction of five compounds were evaluated adequately. The manuscript is interesting and to be overall well written. Not found any critical comments.

There are several typos, for example, in line 210 (E, D -> E, F), line 215 (aan -> van) , and so on. Please check the manuscript thoroughly.

The values of distance in Table 2 to 4 have 6 significant digitals. Reviewer recommends these values are rounded to 3 to 4 significant digitals.

Author Response

Reviewer #2:

Comment

Response

There are several typos, for example, in line 210 (E, D -> E, F), line 215 (aan -> van) , and so on. Please check the manuscript thoroughly.

Correction done and highlighted in yellow.

The values of distance in Table 2 to 4 have 6 significant digitals. Reviewer recommends these values are rounded to 3 to 4 significant digitals.

As suggested by the reviewer, the values in Tables 2-4 have been update to 3 significant digits.

Reviewer 3 Report

In this manuscript, the authors report the isolation and antimicrobial study of five isocoumarines. Compounds were isolated from a fungus Penicillium chrysogenum from Saudi Arabia sediments.  The structures of the 5 isolated isocoumarins are already published, MS and NMR data are compared and appear to be in agreement with literature data. In vitro antibacterial activities are determined on five strains with the highest activities on S. aureus and B. licheniformis. The five isocoumarins are then tested in silico for their interaction with several metallo-b-lactamases by molecular modelling studies.

Comments:

As a general comment, the overall English writing of the manuscript must be largely improved. All parts are concerned.

Another general comment comes from the Introduction and Molecular modelling parts which might be notably reduced and condensed. For example, Tables 2 to 5 could be given as supplementary materials.

Many studies on Isocoumarines have been reported in the past litterature. Several references are given in the References section. Nevertheless, many references, even if relevant, are quite old. Several other more recent references might also be discussed such as, for example: Sila AAD et al. Nat Prod Res 2021 or Xu Z et al Nat Prod Res 2020.

The structures of the isolated isocoumarins show one or two asymmetric carbons which lead to the possibility of enantiomers and diastereoisomers. This point is never discussed in the manuscript. Nevertheless, such determination appears to be essential to be addressed, especially for the consistency of the molecular modelling studies.

Author Response

Reviewer #3:

Comment

Response

As a general comment, the overall English writing of the manuscript must be largely improved. All parts are concerned.

Manuscript been revised and edited in red color by proof readers.

Another general comment comes from the Introduction and Molecular modelling parts which might be notably reduced and condensed. For example, Tables 2 to 5 could be given as supplementary materials.

As per the suggestion, the introduction part was reduced and tables 2-4 have been removed from the main manuscript and placed as Supplementary data (Supplementary Tables 1-3).

Many studies on Isocoumarines have been reported in the past literature. Several references are given in the References section. Nevertheless, many references, even if relevant, are quite old. Several other more recent references might also be discussed such as, for example: Sila AAD et al. Nat Prod Res 2021 or Xu Z et al Nat Prod Res 2020.

As per the suggestion, the reference list was reduced and updated by new references.

The structures of the isolated isocoumarins show one or two asymmetric carbons which lead to the possibility of enantiomers and diastereoisomers. This point is never discussed in the manuscript. Nevertheless, such determination appears to be essential to be addressed, especially for the consistency of the molecular modelling studies.

These are pure isomers as we compared [a]D25 values with the previously isolated ones.

Mellein is not an isocoumarin but a dihydroisocoumarine. The functionnal group present is inn fact a lactone.

Corrected

Round 2

Reviewer 1 Report

Optical rotation can’t give the absolute configuration. If the optical rotation is consistent with reference, the absolution is same. And you should show the absolute configuration in the structures of Figure 1.

Author Response

Optical rotation can’t give the absolute configuration. If the optical rotation is consistent with reference, the absolution is same. And you should show the absolute configuration in the structures of Figure 1.

We matched the Optical rotation and NMR values with the literature and mentioned the absolute configuration of the compounds.

Reviewer 3 Report

In the revised version, the authors have taken into consideration all comments.

Author Response

In the revised version, the authors have taken into consideration all comments.

Thank you so much.